# Water Saving and Yield of Potatoes under Partial Root-Zone Drying Drip Irrigation Technique: Field and Modelling Study Using SALTMED Model in Saudi Arabia

**Abdulrasoul Al-Omran [1,*]** **, Ibrahim Louki [1], Arafat Alkhasha [1,2],**
**Mohamed Hassan Abd El-Wahed [3] and Abdullah Obadi [4]**

[1]  Soil Science Department, College of Food and Agricultural Sciences, King Saud University, Riyadh 11451, Saudi Arabia; ibrahim.louki@hotmail.com (I.L.); aakhasha@ksu.edu.sa (A.A.)

[2]  General Authority for Agricultural Research, El-Kod Station, Aden 1837, Yemen

[3]  Arid Land Agriculture Department, Faculty of Meteorology, Environment and Arid Land Agriculture, King Abdulaziz University, Jeddah 80208, Saudi Arabia; mhassan1@kau.edu.sa

[4]  Department of Plant Production, College of Food and Agricultural Sciences, King Saud University, Riyadh 11451, Saudi Arabia; obadi201@gmail.com

\*  Correspondence: rasoul@ksu.edu.sa; Tel.: +966-11467844; Fax: +966-114678440

**Abstract:** This study aims to evaluate the Partial Root Zone Drying Irrigation System (PRD) as one of the modern technologies that provide irrigation water and increase the efficiency of its use on potato crop. The effect of applying the PRD conventional deficit irrigation (CDI) on the efficiency and water saving in potato crops using the drip surface (S) and subsurface (SS) irrigation methods were investigated. SALTMED model used to predict soil moisture and salinity distribution, soil nitrogen dynamics, and yield of potato crop using the different irrigation levels (150%, 100%, and 50% of Crop evapotranspiration (ETc)). The study showed that the water use efficiency (WUE) decreases with increasing levels of irrigation water, as it ranged between 2.96 and 8.38 kgm$^{-3}$, 2.77 and 7.01 kgm$^{-3}$ for surface irrigation PRD and CDI, respectively, when the amounts of irrigation water varied from 308 mm to 1174 mm, respectively. The study showed that the irrigation efficiencies were the highest when using PRD system in all treatments when irrigating the potato crop during the spring season, and it was more efficient in the case of using subsurface irrigation method. The results show that the soil moisture (SM) was high in 25–45 cm at 150% of ETc was 0.166 and 0.263 m$^3$m$^{-3}$ for the first and last stages of growth, respectively. 100% of ETc, (SM) was 0.296 m$^3$m$^{-3}$ at 0–25 cm, 0.195 m$^3$m$^{-3}$ at 25–45 cm, 0.179 m$^3$m$^{-3}$ at 45–62 cm, depths, respectively. whereas 50% of ETc, (SM) was 0.162 m$^3$m$^{-3}$ at 0–25 cm, 0.195 m$^3$m$^{-3}$ at 25–85 cm, depths. At 100% of ETc, soil salinity was 5.15, 4.37, 3.3, and 4.5 dSm$^{-1}$, whereas at 50%, ETc, these values were 5.64, 9.6, 3.3, and 4.2 dSm$^{-1}$. Statistical indicators showed that the model underestimated yield, for 150%, 100%, and 50% of ETc. Therefore, it can be concluded that yield and WUE using PRD systems were the highest in the potato crop compare to CDI surface and sub-surface, and SALTMED model can predict the moisture distribution, salinity, and yield of potatoes after accurate adjustment.

**Keywords:** partial root-zone drying; SALTMED model; potato crop; irrigation levels

## 1. Introduction

Irrigation is the most critical factor for agricultural production, especially in areas with limited water resources and low annual rainfall, such as arid regions. Saudi Arabia is located in the arid

region; thus, it is critical to consider the quantities of irrigation water and methods of application for any irrigation water management practices. Therefore, modern irrigation systems, such as drip irrigation combined with deficit irrigation (DI) or partial root-zone drying (PRD) system techniques could be useful for saving water. Applying both DI and PRD resulted in the conservation of large quantities of irrigation water and increased yield. Furthermore, using mathematical models helped save time for decision-makers to manage irrigation water and forecast production under different conditions [1], and to study other factors affecting yield, such as soil moisture, salt distribution, and nitrogen concentration in the soil profile.

*1.1. PRD Irrigation Technique*

PRD is an irrigation technique that stimulates partial stomatal closing to decease transpiration into leaves, improves water use efficiency, and increases the yield. PRD includes processes of alternative wetting and drying of both sides of the plant zone to optimize the production of root-sourced chemical signals, which is related to water deficits [2–5]. The PRD technique stimulates some responses associated with drying soil, such as reduced energy and stomatal conductance (gs) while preserving adequate water supply within the wetted part of the root zone to maintain conventional crop growth [6]. This technique of irrigation is very important mainly in arid regions such Saudi Arabia as water resources are very limited. The irrigation water using PRD can be conserved and saved until 50% evapotranspiration without significant reductions in yield.

*1.2. SALTMED Model*

The SALTMED model includes parameters, such as crop evapotranspiration, water uptake, and solute transport under different irrigation systems, drainage, and the relationship between crop yield and water use [7]. Marwa et al. (2020) [8] reported that the SALTMED model is efficient for predicting total dry matter and yield. The SALTMED model can run with different scenarios under different conditions and crop parameters to evaluate the future impact on irrigation management and predict water distribution under automatic irrigation scheduling. Alkhasha and Al-Omran (2019) [9] highlighted that the SALTMED model is reliable for predicting soil moisture and salinity distribution of tomato yield. Kaya and Yazar (2016) [10] concluded that the SALTMED model can be used to compare the simulated and measured soil water content. They found significant yield decreased in 2010, but slightly reduced the high salinity of water from 10–30 dSm$^{-1}$ compared with non-saline water in 2012. The results showed that the model can predict soil water, grain, and dry matter yield of quinoa with a deficit irrigation regime using different water qualities [11].

Kaya et al. (2015) [11] highlighted that the SALTMED model can simulate high relation between soil moisture, total dry matter, and grain yield for quinoa in different irrigation arid environments. The SALTMED model could accurately predict the distribution in the soil profile of salt and soil water content of different crop yields grown under several irrigation demands and environmental conditions. Pulvento et al. (2013) [12] reported that the SALTMED model can predict quinoa crop in Italy under saline and freshwater conditions with high relation between the observed and simulated soil moisture and yield. Abdelraouf and Ragab (2017) [13] concluded that using the model could have reliable results for soil moisture and nitrogen dynamics. Abdelraouf and Ragab (2018) [14] reported that simulating total dry matter, yield, and water yield using the SALTMED model gave good results between the observation and simulation during two seasons (2015 and 2016), with $R^2 = 0.99$. The SALTMED model used in Syria by Gawad et al. (2005) [15] showed the impact of the irrigation method on the soil class, the salinity of irrigation water on soil moisture, and salinity distribution. The results demonstrated that the SALTMED model is useful in the management of water, crops, and soil under field conditions. Ragab et al. (2005) [1] reported that the relationship between both yield and water uptakes as a function of water salinity was nonlinear and defined by a polynomial function of the fourth-order. The relative yield and water uptake obtained by dividing the estimated values by equal values obtained using 100% freshwater alleviated the effect of external factors and produced

consistent and reliable results. Studies conducted by Hirich et al. (2012) [16] and Silva et al. (2013) [17] concluded that SALTMED could make simulations in daily basis according to the main processes of the soil–water–plant relationship. Recently, the model was used on sweetcorn, quinoa, and chickpea and could effectively simulate final yield, moisture, and nitrogen profiles [8,18–20]. It is an important means in scheduling irrigation in a scientifically documented manner with the aim of saving water consumptions and providing an opportunity for horizontal expansion in agriculture by exploiting the limited quantities of water supplied with it, and setting priorities in the use of limited irrigation water.

Therefore, the aim of the study is to introduce all water saving programs such as deficit irrigation and PRD to farmers and examines the SALTMED model to predict soil moisture and soil salinity distributions, soil nitrogen dynamics, and yield of potato crops using the PRD irrigation technique.

## 2. Materials and Methods

### 2.1. Location

This study was conducted from 2014–2018 in the agricultural project in Thadeq Governorate, Central of Saudi Arabia, located between 25°09′16.6″ N and 45°52.3′85″ E). Tables 1 and 2 show the properties (chemical and physical) of soil and irrigation water of the location used in field experiments.

**Table 1.** Properties of soil and water.

| Location | Sand% | Silt% | Clay% | Soil Texture | Bulk Density $gcm^{-3}$ | O.M% | $CaCo_3$% | S.P% |
|---|---|---|---|---|---|---|---|---|
| 1 | 75 | 15 | 10 | Sandy Loam | 1.51 | 1.1 | 18.8 | 26 |
| 2 | 80 | 7.5 | 12.5 | Sandy Loam | 1.56 | 0.9 | 19.9 | 24 |

**Table 2.** Chemical properties of soil and water.

| Sample | pH | E.C | Cations (meq $L^{-1}$) | | | | Anions (meq $L^{-1}$) | | | SAR |
|---|---|---|---|---|---|---|---|---|---|---|
| | | dS·m$^{-1}$ | $Ca^{+2}$ | $Mg^{+2}$ | $Na^{+1}$ | $K^{+1}$ | $Cl^{-1}$ | $HCO_3^{-1}$ | $SO_4^{-2}$ | |
| Location Soil (1) | 7.27 | 13.61 | 55.2 | 22.5 | 35.78 | 12.72 | 53 | 9.9 | 63.7 | 5.74 |
| Location Soil (2) | 7.49 | 3.74 | 17.4 | 10.3 | 7.52 | 3.03 | 12.5 | 3.5 | 22.25 | 2.02 |
| Irrigation Water | 7.6 | 1.6 | 6.55 | 5 | 4.33 | 0.17 | 5.46 | 3.49 | 6.21 | 1.8 |

### 2.2. Climate

The area has hot, dry continental weather in summer and is cold in winter. Temperatures reach above 50 °C in summer, with an average of 44.9 °C [21] and can drop to below −4.4 °C in winter, with an average of 8 °C. The rain season is in the winter but irregular, averaging 101 mm per year [21]. Table 3 illustrates the average climate conditions dominating in the study area from 1998–2018.

**Table 3.** The average of climate conditions in the study area 1998–2018.

| Months | Temperature °C | | Relative Humidity % | | Wind Speed at 2m ms⁻¹ | Evaporation mm | Soil Temperature °C | Radiation Langley day⁻¹ | Hour of Sunshine H day⁻¹ |
|---|---|---|---|---|---|---|---|---|---|
| | Max. | Min. | Max. | Min. | | | | | |
| January | 20.2 | 7.2 | 67 | 25 | 2.7 | 3.8 | 17 | 226 | 6.7 |
| February | 23.8 | 9.5 | 55 | 23 | 3.2 | 5.7 | 17.7 | 306 | 7.5 |
| March | 29.3 | 13.7 | 49 | 18 | 3.2 | 7.6 | 20.4 | 346 | 7.4 |
| April | 34.9 | 19.1 | 47 | 17 | 3.5 | 10.1 | 25.5 | 391 | 7.8 |
| May | 40.3 | 24.1 | 35 | 14 | 3.5 | 13.0 | 30.0 | 422 | 8.5 |
| June | 43.3 | 25.9 | 26 | 12 | 3 | 14.5 | 32.1 | 468 | 10.2 |
| July | 44.2 | 27.6 | 25 | 11 | 3 | 14.5 | 34.0 | 451 | 10.0 |
| August | 44.6 | 27.4 | 30 | 13 | 2.8 | 13.6 | 34.4 | 437 | 10.2 |
| September | 41.1 | 23.6 | 32 | 14 | 2.5 | 11.1 | 32.8 | 396 | 9.6 |
| October | 36.0 | 18.2 | 42 | 17 | 2.1 | 8.5 | 28.1 | 345 | 8.7 |
| November | 28.1 | 13.1 | 68 | 25 | 2.4 | 5.3 | 20.6 | 272 | 7.1 |
| December | 23 | 8.2 | 62 | 22 | 2.7 | 3.7 | 16.2 | 227 | 6.5 |

## 2.3. Field Experiment

The partial dry root-zone irrigation system (PRD) and deficit irrigation for potato experiments were carried out in the open field under surface and subsurface drip irrigation. The number of experimental units are 48 (for 16 treatments and 3 replicates), half of which are for Conventional Drip Irrigation (CDI), and the other half is a Partial Root-zone Drying Irrigation System (PRD). Each treatment is divided into surface drip irrigation (S) and Subsurface drip irrigation (SS). The area of each experimental unit in the open field is 12.5 m² as presented in Figure 1.

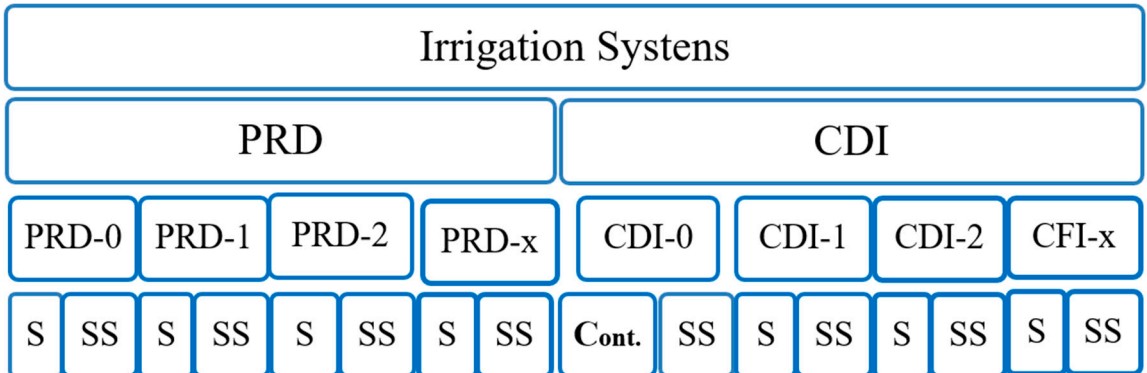

**Figure 1.** Layout of the experimental design for potato open field. PRD = Irrigation with a partial drying of the root zone with two double lines, with four irrigation levels. CDI = Single-line drip irrigation system, with four levels of irrigation: regular, incomplete and conventional. CFI = full drip irrigation or high irrigation. 0, 1, 2, x, = irrigation treatments at 100%, 75%, 50%, and 150% of the planned irrigation water, respectively. Control: Irrigated with conventional standard irrigation (100% conventional surface drip irrigation). S = drip irrigation. SS = subsurface drip irrigation.

Seed potatoes of the Dutch variety "Espunta" (Solanum tuberosum, NAK-NEDERLAND, the Netherlands) were planted, two seeds were planted in each seed hole and thinned to one after its germination ending up to an average of 4 plants per square meter. The experiments were carried out during (October 2014–May 2017).

## 2.4. Fertilization

Basic fertilization was carried out with granulated compound fertilizer with a formula (12-12-17 + 2 Mg + 6 Ca + TE) with a rate of 72 kg/ha-total nitrogen (N), 72 kg/ha-phosphate ($P_2O_5$), and 102 kg/ha- Potassium ($K_2O$), was mixed with soil while preparing it for planting as a basis fertilizer. With the beginning of the third week of planting (the beginning of the second phase), the Fertigation program was implemented with dissolved fertilizers according to the growth stage. Soluble fertilizers were used in the formulation: 1:0.8:3:1.2:0.4:0.4:0.2 and at the rate of 25, 20, 75, 30, 10, 10 and 5 kg/ha of elements (NPK + Ca + Mg + Fe + TE), respectively, weekly until Two weeks before the end of the season, when fertilizing was stopped permanently. In addition, humic acid was added 6% at a rate of 4 L/ha to the mixture in the fertigation tank. Calcium compounds were not mixed with the rest of the fertilizers for fear of sedimentation and clogging of the drippers. The drippers were cleaned by adding 93% phosphoric acid at a rate of 2 L/ha (at a concentration of 0.06 g/L) to the tank on an irrigation day without fertilization.

## 2.5. Daily Readings of Plant Environment Data

The readings of climatic data devices were recorded in the open field daily, as well as readings of the evaporation pan, then using the computer to estimate the daily water needs for irrigation. The water meters were read and recorded before and after each irrigation, and compared to the total volume set for irrigation. The readings of the meters installed in the irrigation lines were recorded

immediately before irrigation and four hours after irrigation. The state of soil moisture in the root zone was also measured periodically by weight method, to compare with estimates of moisture monitoring and measuring soil sensors devices, where samples were taken from each treatment before and after irrigation at a depth of 20 cm and were immediately weighed with a digital scale and then oven drying at 105 °C.

*2.6. Crop Water Requirement*

The crop Evapotranspiration (ETc) in the open field conditions was assessed through three methods namely: Lysimeters, Evaporation Pan, and on Penman-Monteith equation (PM).

2.6.1. Lysimeters

Eight groups of non-weighted lysimeters were prepared in the main crop fields with three equal replications for the potatoes and alfalfa as reference with total of 24 experimental units. The lysimeters of the galvanized sheet lined are equipped with a thermal insulator with dimensions of $3.93 \times 1 \times 1$ m at site No. 1, and with dimensions of $2 \times 1 \times 1$ m in the open field. Each lysimeter was filled with fine gravel at a height of 15 cm [22], and then added alluvial sandy soil similar to that of the experimental fields (Figure 2).

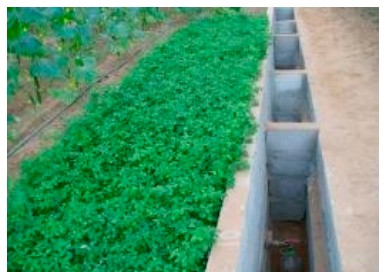 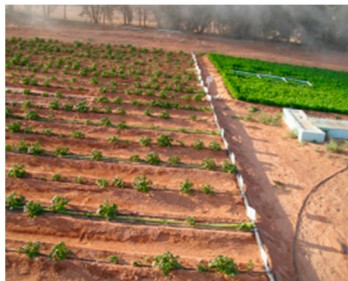 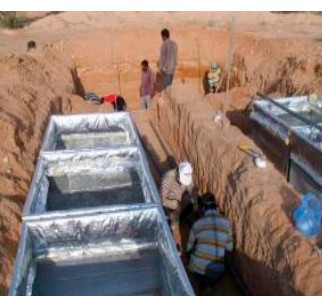

**Figure 2.** The installation of the lysimeters to determine Crop evapotranspiration (ETc.)

The ETc of potato crops was calculated by directly calculating the quantities of ETc from their lysimeters by applying the water balance formula [23]:

$$ETc = P + I - DP \pm \Delta SW \tag{1}$$

where:

ETc = Crop evapotranspiration—crop (in mm) over a period of time.
I = depth of water added by irrigation during a specified time period in mm.
P = amount of rainfall during the same time period in mm.
DP = amount of drainage (in mm) during the same time period.
$\Delta$SW = change in soil moisture content over the same time period in mm.

The value of the crop coefficient (Kc) was calculated from crop evapotranspiration (ETc) and reference evapotranspiration of alfalfa ETr directly from the lysimeters (Table 4). In addition, Kc values were calculated by the evaporation pan method (E-pan) as well as from the Penman-Monteith (PM) equation. The crop coefficient for potato and for the reference alfalfa was calculated according to [23] Table 4. Means of total irrigation water added per season of potato crop during the lysimeter experiments (2014–2017) are shown in Table 5.

**Table 4.** Mean Kc for the main growth stages during the spring and fall seasons taking the alfalfa as a reference for comparison.

| Growth Stage | No. of Days after Planting | Kc—Evap. Pan | Kc-PM Methods | Kc—Alfalfa |
|---|---|---|---|---|
| Spring Season | | | | |
| Kc (ini) early | 1 | 0.69 | 0.55 | 0.50 |
| Kc(ini) end | 20 | 0.69 | 0.55 | 0.50 |
| Kc (mid) early growth season | 47 | 1.00 | 1.16 | 0.91 |
| Kc (mid) early late growth season | 83 | 1.44 | 1.16 | 0.91 |
| Kc end of season | 98 | 1.37 | 1.05 | 0.79 |
| Fall season | | | | |
| Kc (ini) early | 1 | 0.69 | 0.76 | 0.52 |
| Kc(ini) end | 20 | 0.69 | 0.76 | 0.52 |
| Kc (mid) early growth season | 47 | 1.20 | 1.41 | 0.93 |
| Kc (mid) early late growth season | 70 | 1.20 | 1.41 | 0.93 |
| Kc end of season | 79 | 0.88 | 1.07 | 0.65 |

**Table 5.** Means of total irrigation water added per season of potato crop during the lysimeter experiments (2014–2017).

| Method of Calculation of ETc | Spring Season | Fall Season |
|---|---|---|
| Actual water added at 100% | 616.0 | 582.2 |
| Lysimeter using reference crop | 740.0 | 642.2 |
| Evaporation Pan | 532.0 | 748.0 |
| Penman-Monteith | 546.0 | 974.0 |
| RMSE | 1.09 | 1.77 |

### 2.6.2. Pan Evaporation

A pan and adequate weather data station equipment were installed to obtain data weather as well as evaporation pan data [23]. Crop evapotranspiration (ETc) was calculated using the following equation: $ETc = Ep \times Kp \times Kc$, where ETc is maximum daily crop ET in mm, Ep standing for the daily evaporation from class A Pan in mm, Kp is the pan coefficient (ranging between 0.70 and 0.88), and Kc is the crop coefficient (ranging between 0.50 and 1.44) depending on growth stages (Table 4). Kp and Kc were found out according to the equations of Allen et al. (1998). The Gross Water Requirement (GWR) was calculated by the following equations: $GWR = ETc/(1 - LR)$, $GWR = Kc \times Eo \times Kp/(1 - LR) \times Ea$, where, GWR is Gross Water Requirement in mm day$^{-1}$, Ea is irrigation efficiency, and LR the leaching requirement. LR was calculated according to Ayers and Westcot (1985): $LR = ECw \times (El/2\ ECe\ max)$ where, ECw is salinity of irrigation water in dSm$^{-1}$, El is leaching efficiency, and ECe max the maximum electrical conductivity of the extracted soil paste for zero yield in dSm$^{-1}$. The calculated LR in this experiment amounted to 0.06.

### 2.6.3. Penman-Monteith

The combined FAO Penman-Monteith method was used to calculate ET$_o$ or using pan evaporation methods.

### 2.7. SALTMED Model

The full process of calibration, validation, and data required to run the SALTMED model are addressed in our previous study [9]. The data required to run the model are related to crop parameters, soil parameters, meterological daily data, and irrigation data as reported by Alkhasha and Alomran (2019) [9].

Model Statistical Analyses

To define agreement data measured and predicted values for treatments, statistical indicators were used. The root mean square error (RMSE) (Equation (2)) value can be calculated as

$$\text{RMSE} = \sqrt{\sum_{i=1}^{n}(Oi - Si)^2/n}, \tag{2}$$

where $Oi$ is the observed value $i$, $Si$ is the simulated value, and $n$ is the number of treatments.

The coefficient of residual mass (CRM) [24] is defined by

$$CRM = \frac{\sum_{i=1}^{n} Oi - \sum_{i=1}^{n} Si}{\sum_{i=1}^{n} Oi}. \tag{3}$$

CRM measures whether the tendency of the value (negative) overestimates or (positive) underestimates the measurements.

The coefficient of determination ($R^2$) is determined by regression analysis between the observed and simulated values using

$$R^2 = \frac{[\sum_{i=1}^{n}(Oi - O_{avg})(Si - S_{avg})]^2}{\sum_{i=1}^{n}(Oi - O_{avg})^2 \sum_{i=1}^{n}(Si - S_{avg})^2}, \tag{4}$$

where $Oi$ is the observed value $i$, $O_{avg}$ is a mean of values, $Si$ is the $g$-simulated value, $S_{avg}$ is a mean of values, and $n$ is the number of treatments.

The mean relative error (MRE) indicates whether the model is underpredicting or overpredicting the observed value.

$$\text{MRE} = \frac{\sum_{i=1}^{n}(S_i - O_i)}{n}, \tag{5}$$

where $Oi$ is the observed value $i$, $Si$ is the simulated value, and $n$ is the number of observed or simulated values.

## 3. Results

### 3.1. Potato Crop Water Requirements

Potato crop water requirements were estimated according to the evaporation pan method as the main method for daily irrigation while the FAO/Penman-Monteith method was calculated for scrutiny and comparison Table 5. The results showed an increase in the irrigation water for the spring season compared to the fall season in all repeated runs of the experiments, as well as an increase in the quantities of water calculated by the Penman-Monteith method over the calculated pan evaporation method for the same season during the spring seasons, while it decreased slightly in the fall seasons. The seasonal reference evapotranspiration calculated by the evaporation pan method during the fall and spring seasons 532 and 748 mm with daily averages of 5.4 and 6.5 mm/day respectively, while the FAO/Penman-Monteith method were 546 and 975 mm with a daily average of 5 and 8.5 mm/day, respectively.

### 3.2. Potato Yield

In this study the yield of 100% surface conventional drip irrigation treatment was taken as a standard basis for comparison of all potato harvest results for all treatments (Table 6a,b) for spring and fall seasons. The results of the statistical analysis using $LSD_{05}$ showed the extent to which the yield of any treatment increased or decreased over the standard treatment (100 of ETc). Table 6a showed that at PRD-SS yield decreased by 5.1% compared to PRD-S treatment with a slight increase

in the conventional subsurface irrigation CDI-SS by 2.6% (Table 6a), but at the level of irrigation of 75% ETc the yield decrease by 6, 17.8, 12.7, and 22% for the PRD-S, PRD-SS, DI-S, and DI-SS irrigation system, respectively. The results showed that the yield of the PRD-S treatment with was lower when the irrigation was reduced by 75% of ETc, while when the irrigation level was reduced to 50%, the percentage decrease in the yield was 14.8, 23.2, 13.9, and 25.8% for the PRD-S, PRD-SS, DI-S, and DI-SS, respectively. The results of fall season showed the same trend with the exception that at 100% ETc CDI gave a slight increase in yield compared to the PRD method (Table 6b).

**Table 6.** (**a**) Yield of potato as affected by PRD, deficit irrigation (DI), and full irrigation (Spring season). (**b**) Yield of potato as affected by PRD, DI, and full irrigation (Fall season).

| **(a)** | | | | | | | | | | |
|---|---|---|---|---|---|---|---|---|---|---|
| ETc Calculated (mm) | Applied Water (mm) | ETc % | Yield PRD-S (Kg/m²) | WUE Kg/m³ | Yield PRD-SS (Kg/m²) | WUE Kg/m³ | Yield CDI-S (Kg/m²) | WUE Kg/m³ | Yield CDI-SS (Kg/m²) | WUE Kg/m³ |
| | | | | | Spring Seasons | | | | | |
| 616 | 783 | 100 | 3.17 | 4.05 | 3.11 | 3.97 | 3.21 | 4.10 | 3.26 | 4.16 |
| 616 | 783 | 100 | 3.44 | 4.40 | 3.32 | 4.24 | 3.59 | 4.48 | 3.62 | 4.62 |
| 616 | 783 | 100 | 3.66 | 4.67 | 3.27 | 4.18 | 3.43 | 4.38 | 3.60 | 4.60 |
| Mean | | | 3.42 [b] | 4.37 | 3.23 [bc] | 4.12 | 3.41 [b] | 4.36 | 3.50 [ab] | 4.47 |
| 462 | 587 | 75 | 3.33 | 5.67 | 2.55 | 4.34 | 2.98 | 4.92 | 2.59 | 4.41 |
| 462 | 587 | 75 | 2.99 | 5.09 | 2.98 | 5.07 | 3.03 | 5.16 | 2.66 | 4.53 |
| 462 | 587 | 75 | 3.33 | 5.67 | 2.89 | 4.92 | 2.90 | 4.94 | 2.72 | 4.63 |
| Mean | | | 3.22 [b] | 5.49 | 2.81 [c] | 4.79 | 2.97 [b,c] | 5.06 | 2.66 [cd] | 4.53 |
| 308 | 391 | 50 | 3.03 | 7.74 | 2.65 | 6.78 | 2.62 | 6.70 | 2.41 | 6.16 |
| 308 | 391 | 50 | 2.82 | 7.21 | 2.57 | 6.57 | 3.27 | 8.36 | 2.43 | 6.21 |
| 308 | 391 | 50 | 2.83 | 7.24 | 2.62 | 6.70 | 2.94 | 7.52 | 2.74 | 7.01 |
| Mean | | | 2.89 [c] | 7.62 | 2.61 [c,d] | 6.67 | 2.94 [b,c] | 7.52 | 2.53 [c,d] | 6.47 |
| 616 | 1174 | 150 | 4.00 | 3.40 | 3.73 | 3.18 | 3.73 | 3.18 | 3.58 | 3.05 |
| 616 | 1174 | 150 | 3.70 | 3.15 | 3.47 | 2.96 | 3.34 | 2.85 | 3.31 | 2.82 |
| 616 | 1174 | 150 | 3.64 | 3.10 | 3.40 | 3.73 | 3.49 | 2.97 | 3.25 | 2.77 |
| mean | | | 3.78 [a,b] | 3.22 | 3.53 [a,b] | 3.01 | 3.59 [a,b] | 3.06 | 3.38 [b] | 2.88 |
| **(b)** | | | | | | | | | | |
| ETc Calculated (mm) | Applied Water (mm) | ETc % | Yield PRD-S (Kg/m²) | WUE Kg/m³ | Yield PRD-SS (Kg/m²) | WUE Kg/m³ | Yield CDI-S (Kg/m²) | WUE Kg/m³ | Yield CDI-SS (Kg/m²) | WUE Kg/m³ |
| | | | | | Spring Seasons | | | | | |
| 485.3 | 616.7 | 100 | 3.18 | 5.16 | 3.05 | 4.95 | 2.89 | 4.69 | 3.27 | 5.31 |
| 485.3 | 616.7 | 100 | 3.18 | 516 | 3.68 | 5.97 | 3.18 | 5.16 | 3.58 | 5.81 |
| 485.3 | 616.7 | 100 | 3.62 | 5.88 | 3.31 | 5.37 | 2.99 | 4.85 | 3.61 | 5.86 |
| Mean | | | 3.33 [a,b] | 5.40 | 3.35 [a,b] | 5.44 | 3.02b | 4.90 | 3.48 [a,b] | 5.65 |
| 364 | 463.2 | 75 | 2.91 | 6.29 | 2.71 | 5.84 | 2.15 | 4.64 | 2.40 | 5.18 |
| 364 | 462.2 | 75 | 2.70 | 5.83 | 2.77 | 5.98 | 2.46 | 5.70 | 2.11 | 4.55 |
| 364 | 462.2 | 75 | 2.83 | 6.11 | 2.63 | 5.68 | 2.21 | 4.77 | 2.23 | 4.81 |
| Mean | | | 2.81 [b,c] | 6.07 | 2.70 [b,c] | 5.83 | 2.30 [b,c] | 4.97 | 2.24 [c] | 4.84 |
| 242.7 | 308.15 | 50 | 2.09 | 6.79 | 2.20 | 7.14 | 1.81 | 5.87 | 1.94 | 6.3 |
| 242.7 | 308.15 | 50 | 1.97 | 6.40 | 2.58 | 8.38 | 1.81 | 5.87 | 1.74 | 5.65 |
| 242.7 | 308.15 | 50 | 1.90 | 6.17 | 2.29 | 7.44 | 1.87 | 6.07 | 1.94 | 6.30 |
| Mean | | | 1.99 [d] | 6.46 | 2.36 [c,d] | 7.66 | 1.81 [d] | 5.87 | 1.87 [d] | 6.07 |
| 485.3 | 924.5 | 150 | 3.23 | 3.49 | 3.84 | 4.15 | 3.86 | 4.17 | 4.12 | 4.46 |
| 485.3 | 924.5 | 150 | 3.33 | 3.60 | 3.77 | 4.08 | 3.50 | 3.78 | 3.17 | 3.43 |
| 485.3 | 924.5 | 150 | 2.98 | 3.22 | 3.12 | 3.37 | 3.29 | 3.56 | 3.41 | 3.69 |
| Mean | | | 3.18 [b] | 3.44 | 3.38 [a,b] | 3.66 | 3.55 [a,b] | 3.84 | 3.57 [a,b] | 3.86 |

Different letters means that values are different at 5% of least significant difference (LSD0.05).

*3.3. SALTMED Data*

The soil water status and salinity distribution and soil nitrogen in the root zone for all irrigation treatments were predicted using the SALTMED model. The data were selected from three irrigation regimes 150%, 100%, and 50% of ETc using the partial root-drying technique. The data represent the prediction values during the four stages of growth of potato crops.

3.3.1. Water Applied at 150% of ETc

Soil Moisture Distribution

Figure 3A(a–d) shows the distributions of soil moisture ($\theta_v$) during the four stages of growth under 150% of ETc. Each appears to have a specific pattern. Soil moisture (SM) contents differ between the growth stages, placement of the dripper, and soil depth. However, SM at first was 0.146, 0.166, and 0.124 $m^3m^{-3}$ as an average at depths of 0–25, 25–45, and 45–120 cm, respectively. The highest value was at subsurface (25–45 cm) under the dripper, whereas at surface (10) cm was less than 0.1 $m^3m^{-3}$ because of the default effect of evaporation. Meanwhile, at the second stage, the soil moisture distribution was semi-elliptical and increased, compared with the first stage, with values of 0.248, 0. 227, 0.195, and 0.101 $m^3m^{-3}$ at soil depths of 0–25, 25–45, 45–100, and 100–120 cm. Furthermore, the $\theta_v$ in the middle stage was distributed more vertically, where the values of SM were 0.274, 0.286, 0.206, and 0.11 $m^3m^{-3}$ in depths of 0–25, 25–45, 45–100, and 100–120 cm, respectively. The SM content was similar throughout the depths, especially at 50 cm horizontally from the plant in the horizontal direction, and these values were 0.275, 0.263, 0.251, and 0.240 $m^3m^{-3}$ in soil depths of 0–25, 25–45, 45–100, and 100–120 cm.

Soil Salinity Distribution

The soil water content predominantly affects the soil salinity distribution presented in Figure 3B(a–d). The salt concentration was relatively high on the surface, especially during the first stage and away from the dripper. On average, salt concentrations were 9.5, 4.2, 3.7, and 2.8 $dSm^{-1}$ in the first stage of growth, and in the second stage, the soil salinity was more obvious at 45 cm soil depth. On both sides of the line, soil salinity was 10.5 and 15.2 $dSm^{-1}$ at 10 cm soil depth. However, soil salinity was decreased to 4.9, 3.3, and 2.5 $dSm^{-1}$ at soil depths of 25–45, 45–100, and 100–120 cm, respectively. The soil salt concentration increased under the dripper with growth stages, especially in the surface layer, with values of 2.4, 3.3, 4.2, and 2.4 $dSm^{-1}$, whereas during the last stage, it increased to 2.6, 2.6, 3.55, and 4.5 $dSm^{-1}$ at soil depths of 0–25, 25–45, 45–100, and 100–120 cm, respectively.

Soil Nitrogen

The SALTMED model showed varying results in predicting the movement of nitrogen by tracking the concentration. However, the higher concentration of nitrogen Figure 3C(a–d) was in the surface layer (0–15 cm) and ranged between 11.7 and 51.5 $mgL^{-1}$, especially in the first growth stage. In the second growth stage, the results of the prediction showed a high concentration of soil nitrogen of 24 $mgL^{-1}$ at 10–25 cm soil depth, whereas subsurface layers showed levels of 13.8 and 12.5 $mgL^{-1}$ at 10 cm and 25–40 cm, respectively. The rest of the soil profile up to 120 cm showed a concentration of 1.5 $mgL^{-1}$ on average. However, during the third and fourth growth stages, the nitrogen movement was more obvious at a soil depth of 120 cm. The highest concentration was around 19.65 $mgL^{-1}$ at 45–65 cm and the distribution was heterogeneous on both sides until 75 cm horizontally. As for the last stage of plant growth, the concentration was different, reaching the lowest levels of ~2.4 $mgL^{-1}$ in the surface layer (0–15 cm), and the concentration increased vertically where the highest concentration was 20.5 $mgL^{-1}$ at 65–100 cm.

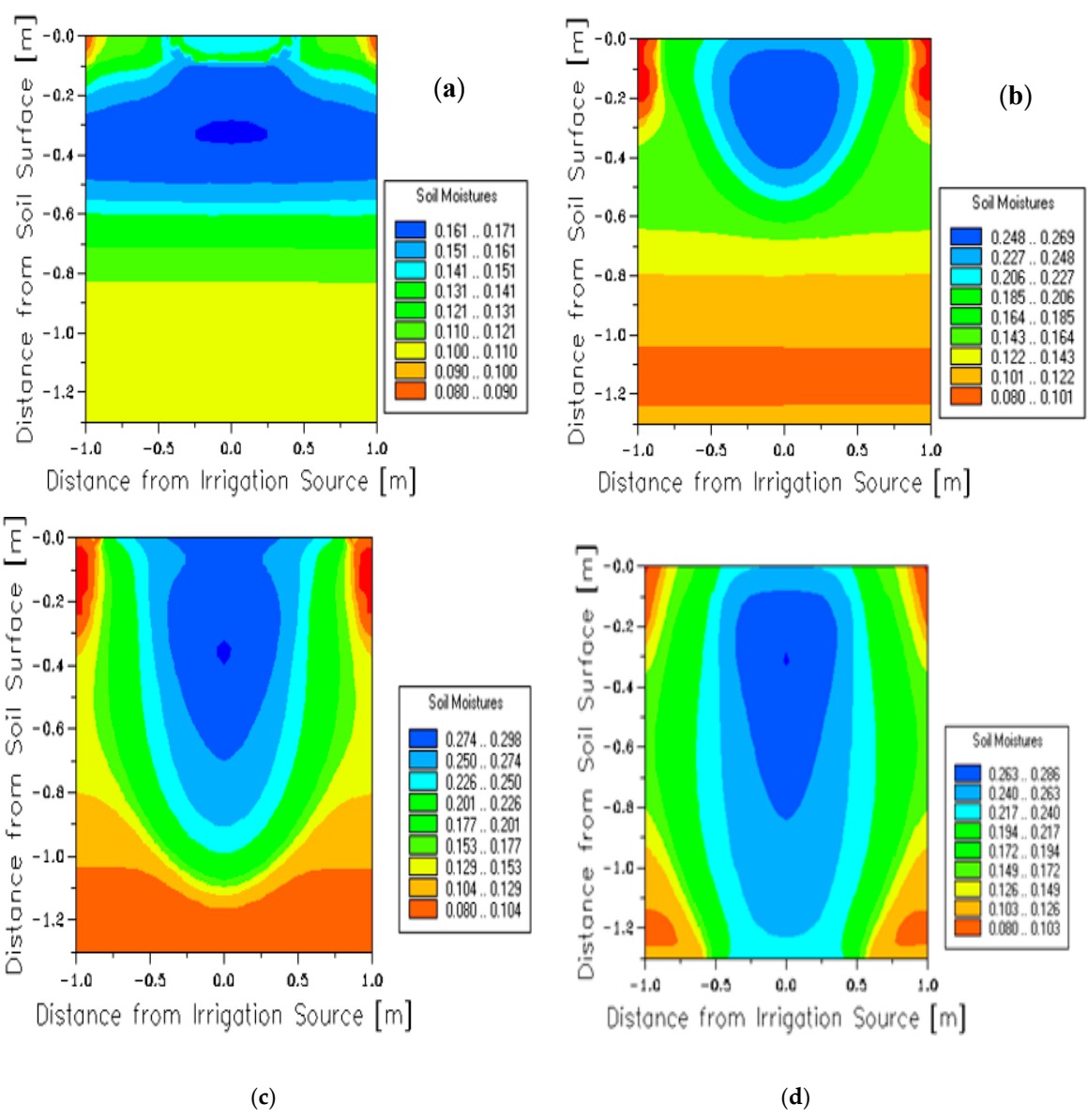

A

**Figure 3.** *Cont.*

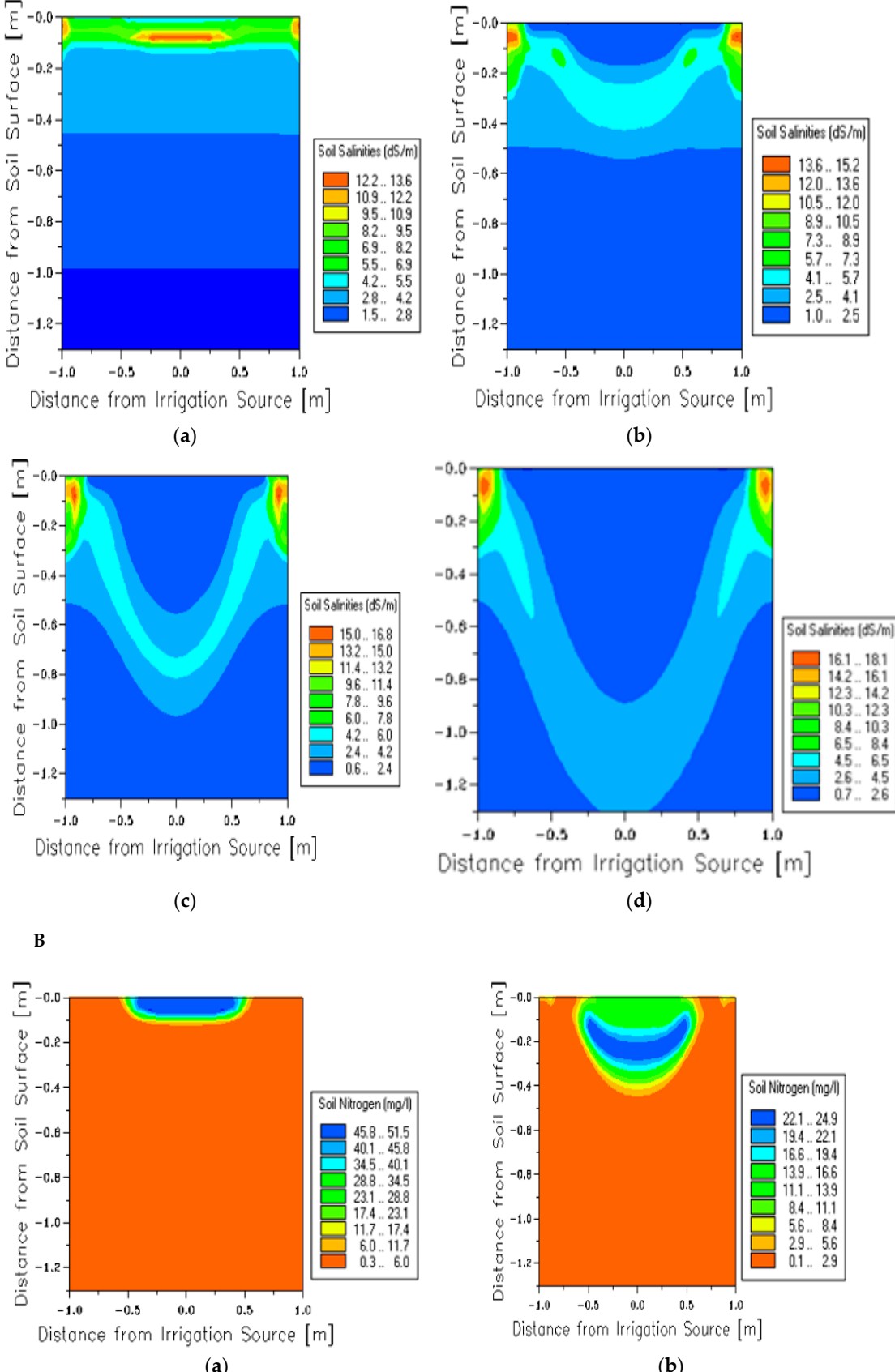

**Figure 3.** *Cont.*

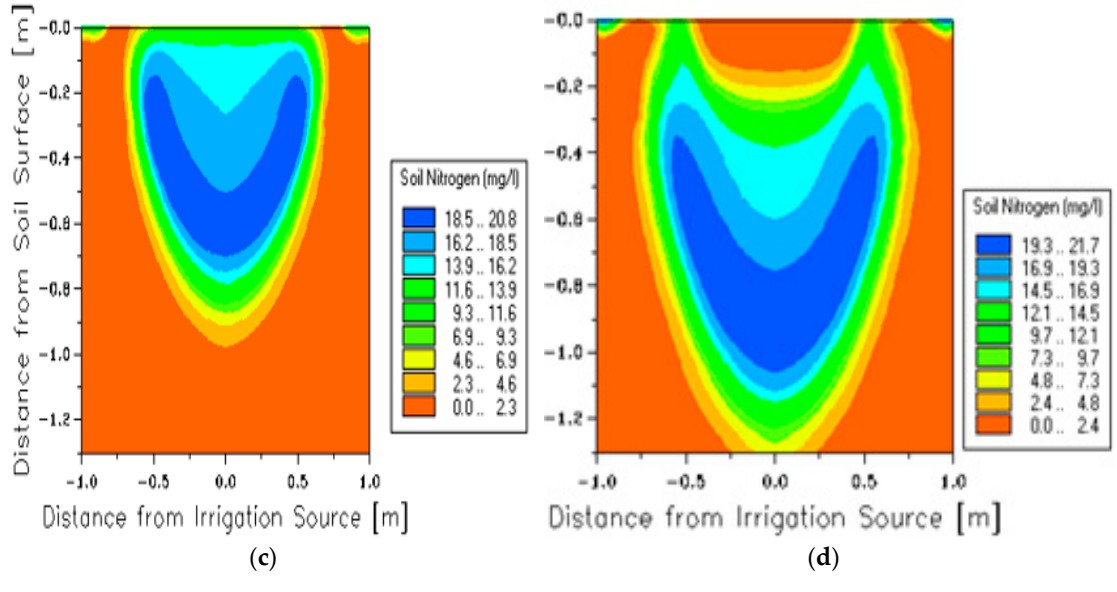

C

**Figure 3. A(a–d)**. The distribution of soil moisture, through stages (**a–d**) of growth in the soil profile under partial root drying for potato crops at 150% of ETc, produced by SALTMED model version 2020. **B(a–d)**. The distribution of soil salinity through stages (**a–d**) of growth in the soil profile under partial root drying for potato crops at 150% of ETc, produced by SALTMED model version 2020. **C(a–d)**. The distribution of soil nitrogen through stages (**a–d**) of growth in the soil profile under partial root drying for potato crops at 150% of ETc, produced by SALTMED model version 2020.

### 3.3.2. Water Applied at 100% of ETc

Soil Moisture Distribution

Figure 4A(a–d) show the SM ($\theta_v$) through all growth stages at 100% of ETc of water applied. Each has a specific pattern. SM contents were different between growth stages and dripper line placements. SM content ($\theta_v$) values were 0.162, 0.179, and 0.122 $m^3m^{-3}$ as an average at depths of 0–25, 25–62, and 63–120 cm, respectively. The highest value was at subsurface (25–62 cm) in the center of the dripper line. In the second stage, the distribution of SM was semi-elliptical from the surface to 57 cm of soil depth, and higher than that in the first stage, with values of 0.242, 0.195, 0.127, and 0.0.091 $m^3m^{-3}$ at 0–25, 25–45, 45–100, and 100–120 cm soil depths, respectively. Furthermore, the $\theta_v$ in the intermediate stage was more distributed vertically, where the values of $\theta_v$ were 0.296, 0.270, 0.194, and 0.118 $m^3m^{-3}$ at depths of 0–25, 25–45, 45–100, and 100–120 cm, respectively. The moisture content was alike on both sides vertically with values of 0.242 and 0.10 $m^3m^{-3}$.

Soil Salinity Distribution

Generally, the salt distribution Figure 4B(a–d) was approximately raised on the surface, specifically during the first stage and away from the plant. The salt concentrations were 9.38, 4.75, 3.7, and 2.8 $dSm^{-1}$ as an average, at soil depths of 0–20, 20–45, and 45–120 cm. In the second stage, the salinity distribution was clear to a soil depth of 43 cm and both sides of the plant were 3.1 and 5.65 $dSm^{-1}$ at 15 cm and 15–43 cm soil depths, respectively. However, salinity during the third and last stages was decreased and more obvious toward the bottom. On the opposite in maximum distance horizontally, the values were 2.7, 5.4, and 1.8 $dSm^{-1}$ for 0–25, 25–60, and 60–120 cm, respectively, in the third stage. Furthermore, the salinity during the last stage of growth was more concentrated horizontally and away from the dripper, and the soil salinity was ~4.5 $dSm^{-1}$ as an average.

Soil Nitrogen

Figure 4C(a–d) shows a part of the results as predicted concentration and movement of soil nitrogen as $NO_3^-$. The results show an increase in the concentration to 33.5 mgL$^{-1}$ in the surface layers (0–15 cm), compared to the rest (15–120 cm) of the soil depths, which was 6.4 mgL$^{-1}$ in the first growth stage. In the second growth stage, the results of the modeling showed that the highest concentration of soil nitrogen was 26 mgL$^{-1}$ at a soil depth of 15–25 cm, contrary to the surface and subsurface layers of 16.3 and 13.06 mgL$^{-1}$, respectively. Furthermore, during the third and fourth growth stages, the nitrogen movement was more obvious to the bottom of the soil profile (120 cm). The highest concentration of 22.2 mgL$^{-1}$ at 20–45 cm and the distribution was heterogeneous on both sides until 80 cm horizontally. As for the last stage of plant growth, the concentration was different, reaching the lowest levels of ~4.6 mgL$^{-1}$ in the surface layer (0–10 cm), and the concentration differed vertically, where the highest concentration was 18.3 mgL$^{-1}$ at a depth of 10–110 cm.

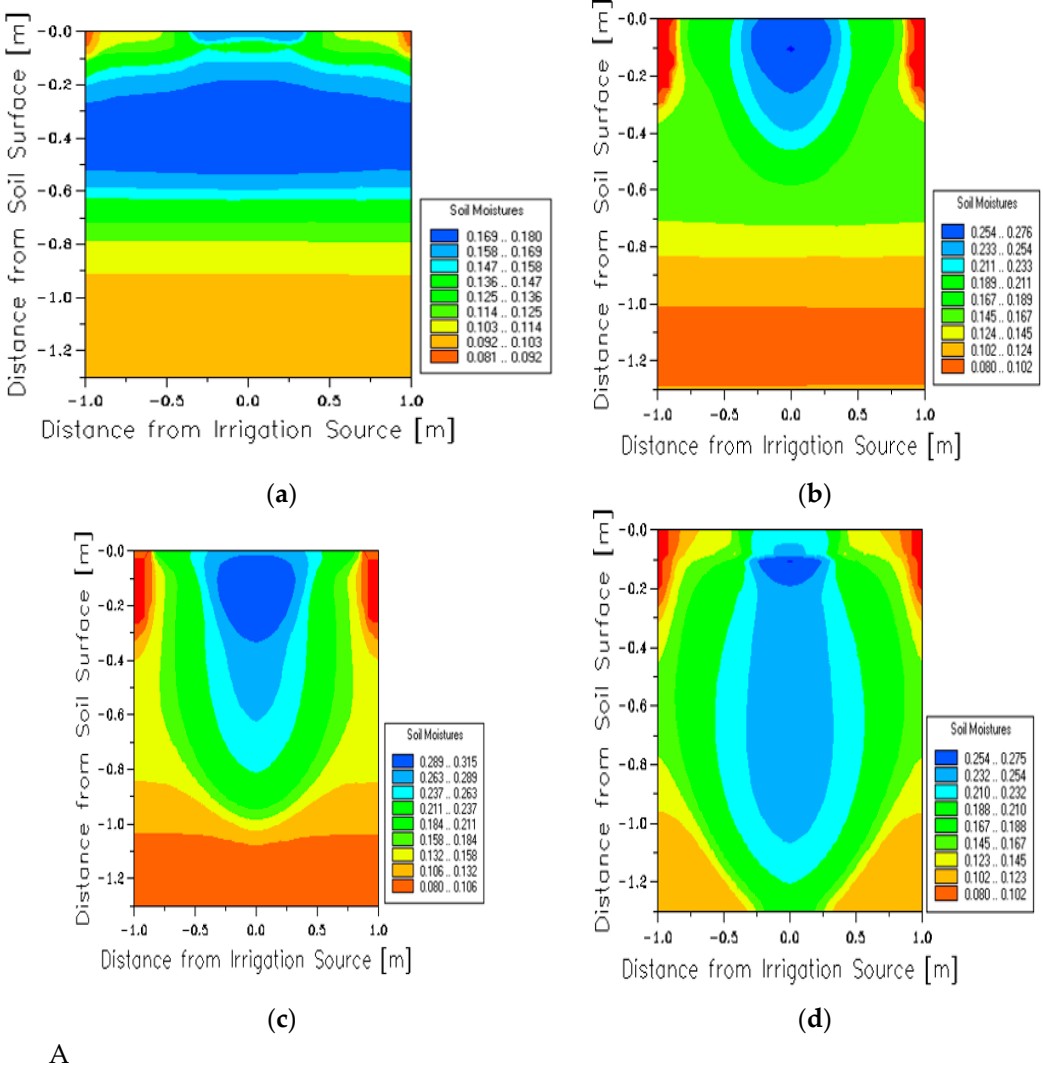

A

**Figure 4.** *Cont.*

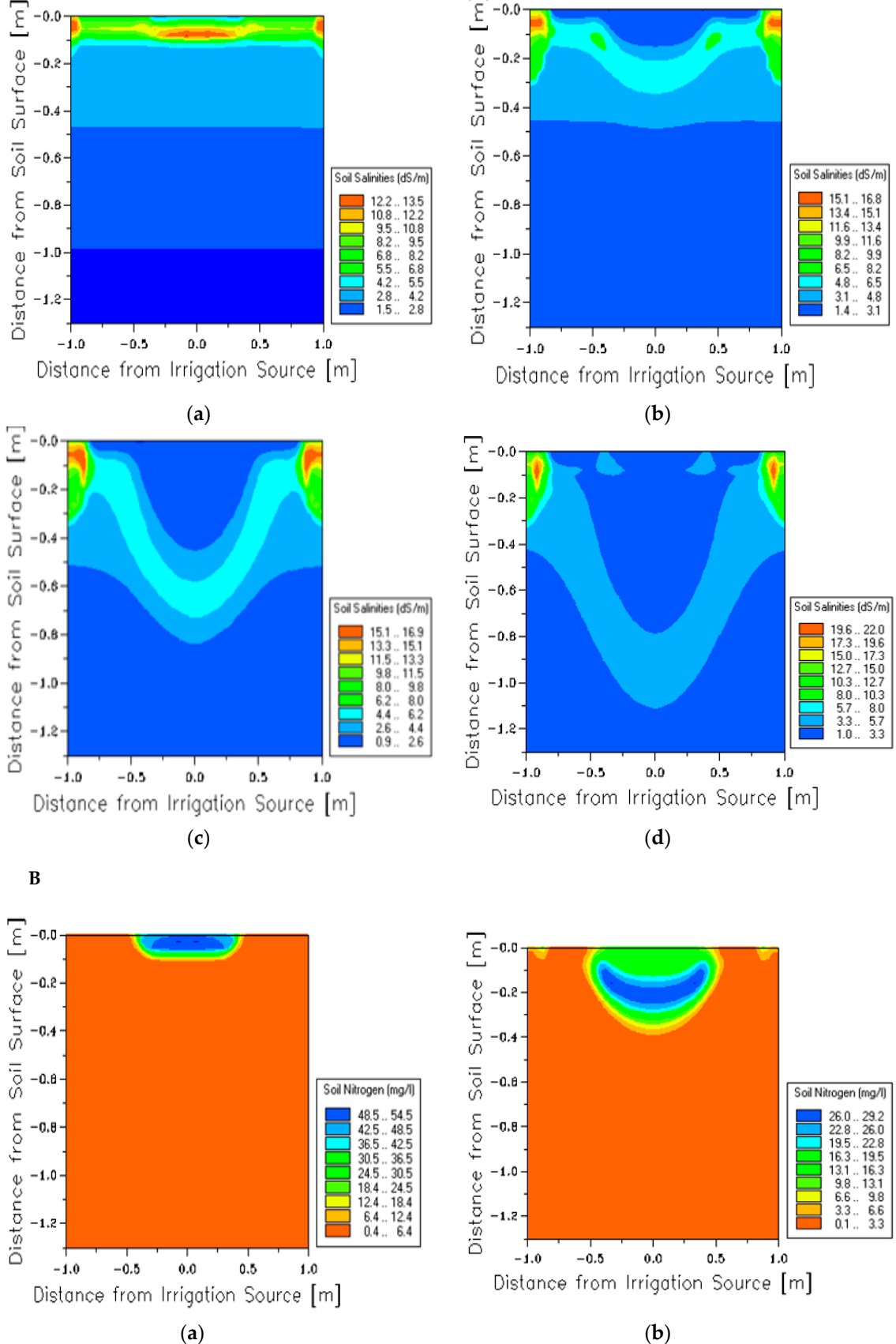

**Figure 4.** *Cont.*

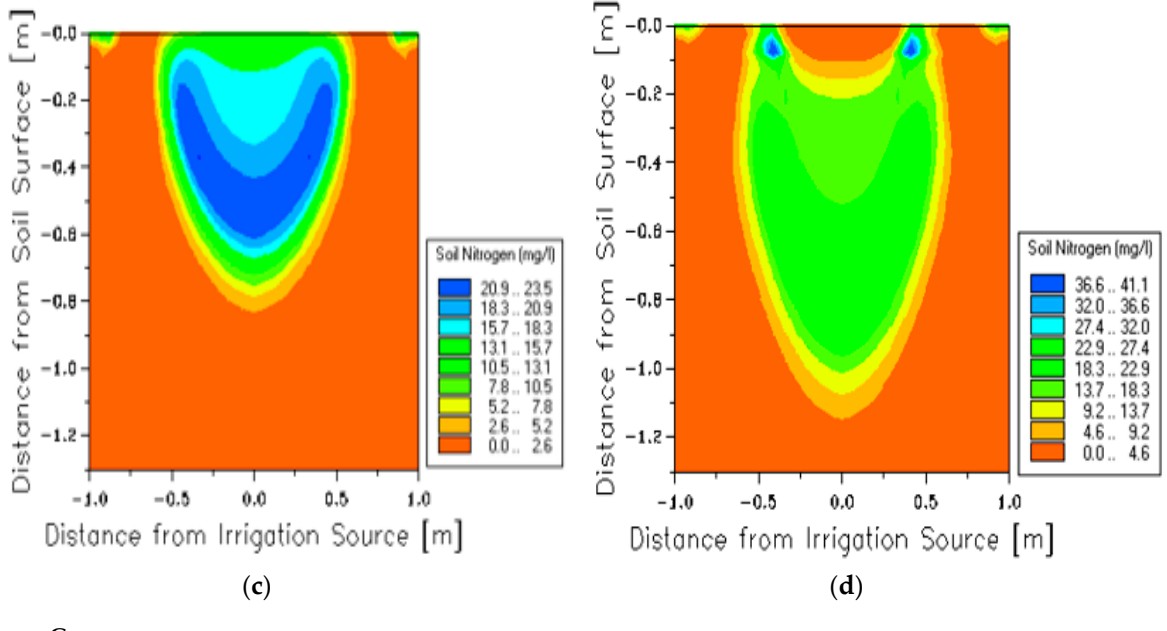

C

**Figure 4. A(a–d).** The distribution soil moisture through stages (**a–d**) of growth in the soil profile under partial root drying for potato crops at 100% of ETc, produced by SALTMED model version 2020. **B(a–d).** The distribution of soil salinity through stages (**a–d**) of growth in the soil profile under partial root drying for potato crops at 100% of ETc, produced by SALTMED model version 2020. **C(a–d).** The distribution of soil nitrogen through stages (**a–d**) of growth in the soil profile under partial root drying for potato crops at 100% of ETc, produced by SALTMED model version 2020.

### 3.3.3. Water Applied at 50% of ETc

Soil Moisture Distribution

Figure 5A(a–d) show different SM distribution patterns under severe irrigation levels of 50% of ETc. However, the $\theta_v$ values as an average of the 0–25, 25–62, 63–100, and 100–120 cm soil depths were 0.162, 0.18, 0.122, and 0.104 m$^3$m$^{-3}$, respectively. The highest moisture content was at 25–63 cm. In the second stage of growth, the moisture distribution was identical around the plant until 40 cm soil depth. The $\theta_v$ was increased by 49.38% in the surface layer compared to the previous stage to 0.242, 0.195, and 0.127 m$^3$m$^{-3}$ for 0–25, 25–85, and 86–120 cm, respectively. However, the $\theta_v$ during the third growth stage was 0.296, 0.270, 0.194, and 0.118 m$^3$m$^{-3}$ for 0–25, 25–60, 61–100, and up to 120 cm, respectively. The soil moisture content was 0.242 m$^3$m$^{-3}$ as an average and was similar along the soil profile, increasing from the top layer to subsurface layers.

Soil Salinity Distribution

Figure 5B(a–d) show the soil salinity distribution, showing the highest percentage accumulation of salt in the surface soil layer as 9.38 dSm$^{-1}$ as an average at 0–20 cm of soil depth during the first stage of growth, whereas the other soil profile salinity was 4.75 and 2.8 dSm$^{-1}$ for 20–45 and 45–120 cm, respectively. In the second stage, the highest salinity concentration predicted was 5.65 dSm$^{-1}$ at 15–43 cm soil depth, compared to the surface layer (0–15 cm) at 3.1 dSm$^{-1}$ and the subsurface layer (43–120 cm) at 2.25 dSm$^{-1}$. The salinity was higher away from the plant horizontally and reached 16.05 dSm$^{-1}$ at 100 cm. Furthermore, during the third and last stages of growth, the salinity concentration moved downward, concurring with the decrease in the surface layer, irrespective of concentrations horizontally. The salinity values were 2.7, 5.4, and 1.8 dSm$^{-1}$ for 0–25, 25–60, and 60–120 cm depths during the third stage, whereas in the last stage, the value was 4.2 dSm$^{-1}$ as an average.

Soil Nitrogen

The decrease in irrigation to 50% ETc affected the nitrogen concentration using the SALTMED model prediction. Figure 5C(a–d) show the prediction concentration of $NO_3^-$ and the average value was 32.85 mgL$^{-1}$ for the first stage, whereas the concentration was decreased with depth. However, as the growth days advanced, the high concentrations increased with depth to 18.63, 15.88, and 20.08 mgL$^{-1}$ for 15–25, 15–60, and 15–85 cm soil depth in the second, third, and last stage of growth, respectively.

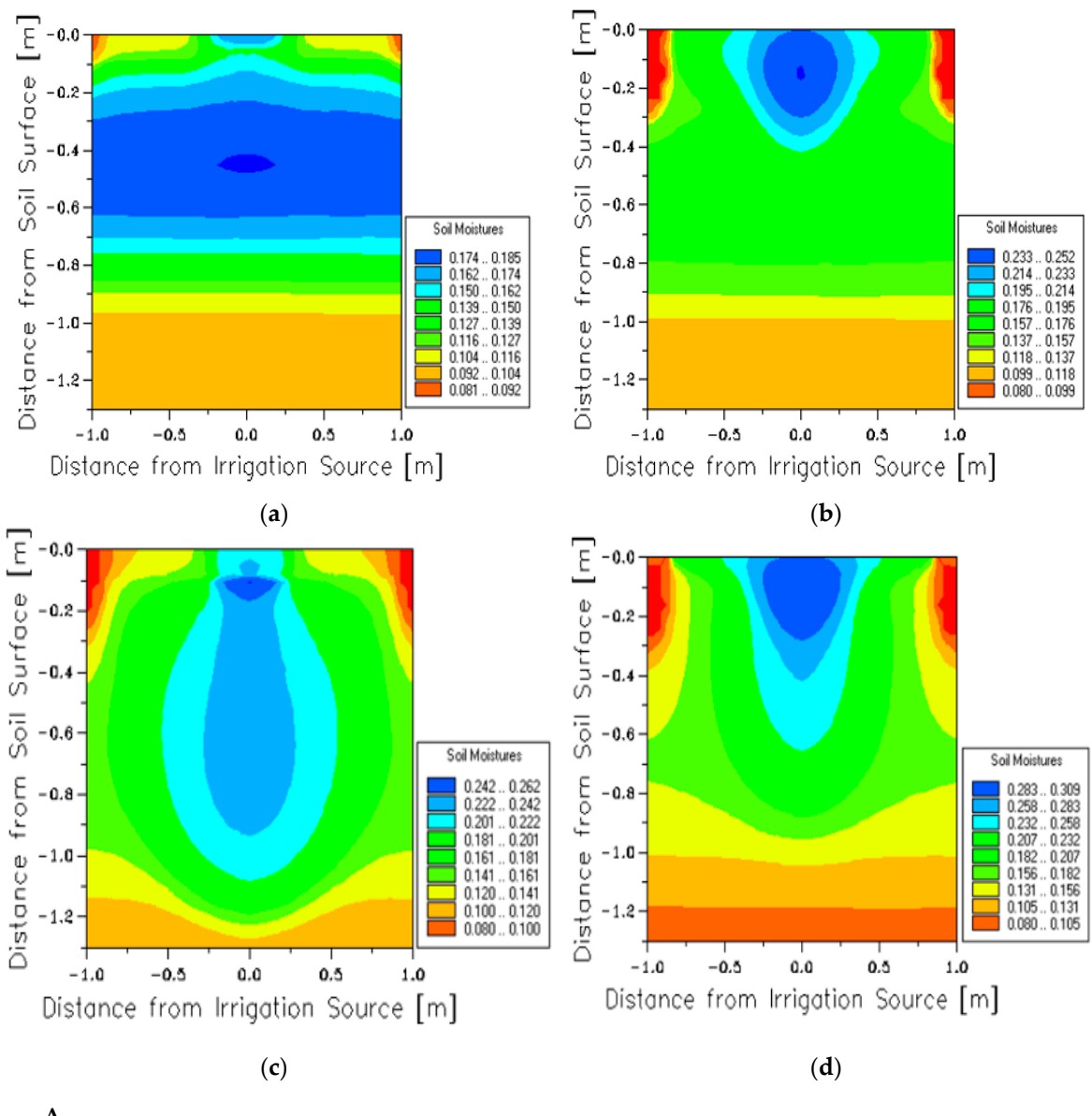

(a)

(b)

(c)

(d)

A

**Figure 5.** *Cont.*

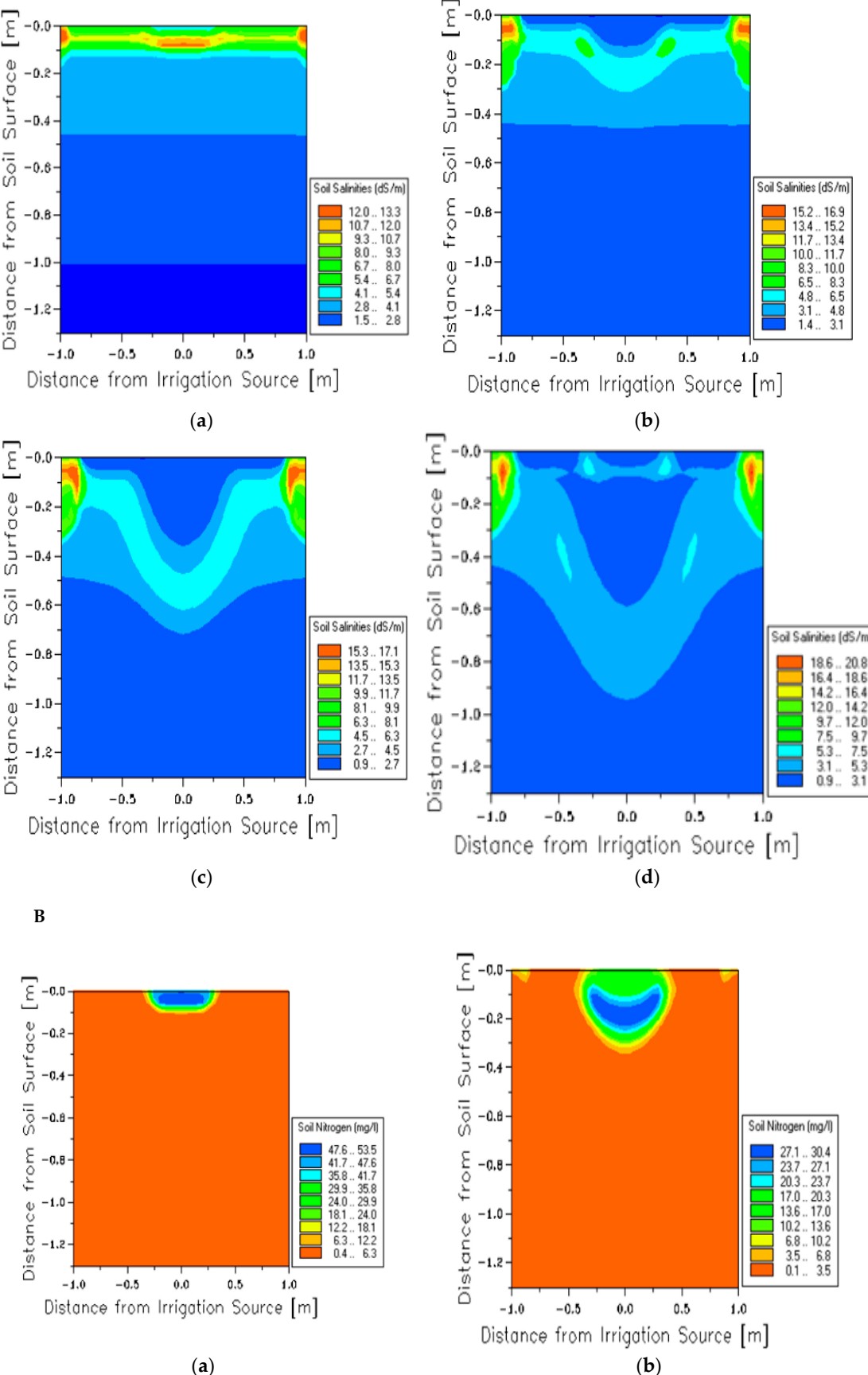

**Figure 5.** *Cont.*

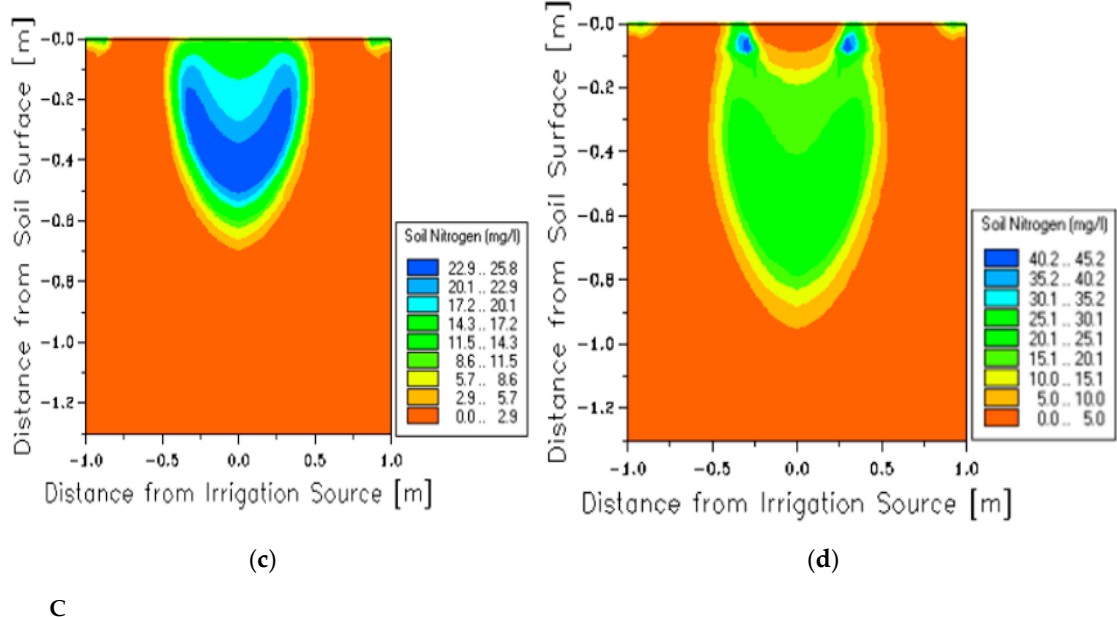

(**c**)                                                                                      (**d**)

C

**Figure 5. A**(**a–d**). The distribution of soil moisture through stages (**a–d**) of growth in the soil profile under partial root drying for potato crops at 50% of ETc, produced by SALTMED model version 2020. **B**(**a–d**). The distribution soil salinity through stages (**a–d**) of growth in the soil profile under partial root drying for potato crops at 50% of ETc, produced by SALTMED model version 2020. **C**(**a–d**). The distribution of soil nitrogen through all stages (**a–d**) of growth in the soil profile under partial root drying for potato crops at 50% of ETc, produced by SALTMED model version 2020.

The Yield

The results in Table 7 show statistical indicators of yield, showing a good correlation between the observed and simulated production of potatoes with PRD, whereas the yield decreased in the predicted compared to the observed data, especially with high deficit irrigation at 50% of ETc. Table 8 compares the observed and simulated yield of potatoes and relative errors (RE) for different water irrigation levels. The RE values ranged from 6.51 to 19.63%. The results of the model are authoritative to simulate potato yield in arid conditions.

**Table 7.** Statistical indicators of soil moisture content, soil salinity, and three irrigation regimes.

| Treatment | MAD | MSE | RSME | MAPE | MRE | CRM | $R^2$ |
|---|---|---|---|---|---|---|---|
| SM | 0.01056 | 0.00016 | 0.01278 | 5.75750 | 0.00011 | −0.00061 | 0.94561 |
| SS | 4.50000 | 22.97829 | 4.79357 | 107.52623 | 4.50000 | −0.79662 | 0.93944 |
| 150 | 3.16551 | 10.69341 | 3.27008 | 8.42691 | −3.16551 | 0.08370 | 0.88020 |
| 100 | 5.26852 | 27.77429 | 5.27013 | 15.40583 | −5.26852 | 0.15374 | 0.99982 |
| 50 | 3.86148 | 16.73179 | 4.09045 | 13.50786 | −3.86148 | 0.13362 | 0.90908 |

**Table 8.** Observed versus simulated total potato yield.

| % of ETc | Observed, t ha$^{-1}$ | Simulated, t ha$^{-1}$ | RE % |
|---|---|---|---|
| 150 | 40.03 | 37.25 | 6.95 |
| | 37.00 | 34.59 | 6.51 |
| | 36.43 | 32.12 | 11.82 |
| 100 | 31.72 | 26.60 | 16.15 |
| | 34.44 | 29.20 | 15.23 |
| | 36.64 | 31.20 | 14.84 |
| 50 | 30.25 | 28.02 | 7.37 |
| | 28.19 | 22.66 | 19.63 |
| | 28.25 | 24.43 | 13.52 |

## 4. Discussion

### 4.1. Crop Water Requirements

The results in Table 5 are relatively consistent with what was reported by Van der Zaag (1991) that the average daily sprinkler irrigation of potato crops in the fall season in the Kingdom during the months of November, December, and January is 12 mm every 3 days at a rate of 4 mm/day and for the spring season during the months of February, March and April is 5 and 10 and 12 mm/day. These values are also in line with what was mentioned by Al-Omran et al. (2019) [25] that the water requirements for irrigation of potatoes in the Riyadh region during the fall and spring season were 6207 and 7536 $m^3$/ha/season (i.e., 620.7 and 753.6 mm), respectively. In the case of drip irrigation with saline water the value was 1000 ppm, which is also within the FAO estimate of 500–700 mm [26].

### 4.2. The Effect of Water Application on Yield during the Spring Season

It was observed that the decrease in yield as a result of a 50% reduction in irrigation in the surface drip irrigation treatment was less than the decrease in subsurface irrigation by almost twice as much in both PRD and DI cases during this season, contrary to what had been expected. By increasing the irrigation percentage to 150%, the yield of the treatments did not increase by that percentage, as the percentage of increase was 11.3%, 4.0%, and 5.5% for the PRD-S, PRD-SS, and CFI-S treatments, respectively, while this increase in irrigation has negatively affected the yield of potatoes in the subsurface irrigation treatment (CFI-SS).

The increase in irrigation rates has led to a decrease in the water use efficiency (WUE) (Table 6a,b). The results showed that WUE by PRD system is higher than conventional full irrigation (CFI) or conventional deficit irrigation (CDI) in all cases. The results indicated that the WUE decreases with increasing levels of irrigation water, as it ranged between 3.10 and 7.74 $kgm^{-3}$, 2.96, and 6.70 $kgm^{-3}$, for surface irrigation PRD and CDI, respectively, when the amounts of irrigation water varied from 308 mm to 1174 mm, respectively, while PRD-SS and CDI-SS irrigation were 2.77–8.38 $kgm^{-3}$, and 2.77–7.01 $Kgm^{-3}$, respectively for the same amounts of irrigation water. The results indicated that in the spring season, the subsurface drip irrigation method was the least efficient in using water, followed by the conventional surface irrigation. The results showed that the water yield was highest when applying the PRD-S system, followed by PRD-SS.

### 4.3. The Effect of Water Application on Yield during the Fall Season

The yield of the CFI-S drip irrigation treatment of 100% was taken as a standard basis for comparison of all potato harvest results for all treatments. Results of statistical analysis using $LSD_{05}$. The results showed that—and unlike the spring season—the yield of potatoes under surface irrigation method gave higher yield than the other treatments with rates of 10.6% and 15.3% with the PRD-SS and CDI-SS, respectively, than the conventional surface CDI-S (Table 6b). However, with a 75% irrigation treatment, the decrease in yield in CFI-S was approximately 24.1 and 25.5% for the CDI-S and CDI-SS irrigation system, respectively, while the decrease in PRD system was 6.7 and 10.4% for the PRD-S and PRD-SS irrigation system, respectively. With the irrigation level reduced to 50% the percentage decrease in yield was 34.1, 22.1, 39.5, and 37.8% for the irrigation treatments PRD-S, PRD-SS, CDI-S and CDI-SS, respectively. It was observed in this study that the percentage of decrease in PRD-SS was the lowest, in contrast to what was observed in the spring season. With an increase in the irrigation ratio to 150%, there was an increase in yield of 18.6, 17.8, 12.2, and 5.3% for the PRD-S, PRD-SS, CFI-S and CFI-SS treatments, respectively. However, this increase does not correspond to the increase in irrigation. The results of the fall season did not differ from the spring season, as the water yield was the highest when the PRD-S system was applied with surface irrigation, followed by subsurface irrigation with the same system. Table 6b illustrated by the statistical analysis, the decrease in water use efficiency (WUE) is more rapid with the increase in the added water when applying the conventional subsurface irrigation system.

### 4.4. SALTMED

After the successful (calibration and validation) processes of SALTMED, it was used for other irrigation treatment data. The agreement between the observed and simulated data for SM and soil salinity through all soil depths (0–120 cm) is shown in Figure 6a,b, and statistical indicators are shown in Table 7. The data show a slight difference existed between the observed and simulated data for moisture and salinity. The results showed that the value of RSME was 0.0128 for 100% of ETc, and $R^2$ was 0.95 for soil moisture for soil depths of 0–120 cm, and CRM was –0.00061 overestimated. However, statistical indicators of soil salinity were 4.78, –0.79662, and 0.94 for RSME, CRM, and $R^2$, respectively. Here, the CRM values were negative, indicating that the model overestimated for moisture and salinity. The surface layer of soil showed less moisture, especially at 50% of ETc. The advanced growth stage could be affected by weather fluctuation, the root of the plant, and most dynamic changes, such as soil evaporation and agricultural processes. Furthermore, in high irrigation regimes at 150% of ETc, the soil moisture was high in depths of 100–120 cm compared with other irrigation levels because of the high added water and texture of sand soil characterizing a high infiltration rate. However, the salt concentration percentage was high in the surface layers far from the source of irrigation because of high evaporation [8,27–29]. Generally, the relationship between observed and simulated values under all irrigation regimes with the PRD technique showed a high correlation, which is a good indicator of the SALTMED model in predicted soil moisture and salinity distribution.

The model was good for the predicted yield of potatoes under all irrigation regimes after calibration processes. Figure 7a–c shows the correlation of the model for yield for 150%, 100%, and 50% of ETc, respectively. The efficiency of the model in predicting yield was measured using the statistical indicators presented in Table 7. The $R^2$ values were 0.88, 0.99, and 0.90, and RSME were 3.27, 5.27, and 4.09 for 150%, 100%, and 50% of ETc, respectively. The CRM values were 0.08 to 0.15, which was slightly underestimated. The RE values in Table 5 between the observed and simulated data ranged from 6.51% to 19.63%. Karandish and Simunek (2019) [30] reported that RE values ranged from 3.5–8.3% in 2010 and 3.6–7.9% in 2011. Hassanli et al. (2016) [31] reported that RE values ranged from 0.9–24.7% with corn crop yield under different quality and saline treatments. Ragab et al. (2005) [1] reported RE values in the range of 0–21.5% as an average of 5.7%, and Razzaghi et al. (2011) [32] highlighted that RE was 0.8–2.2% as an average of 1.5% for quinoa crop. The data indicated that the SALTMED model perfectly estimated potato yield. These results agree with many other studies [8,9,18–20,33,34].

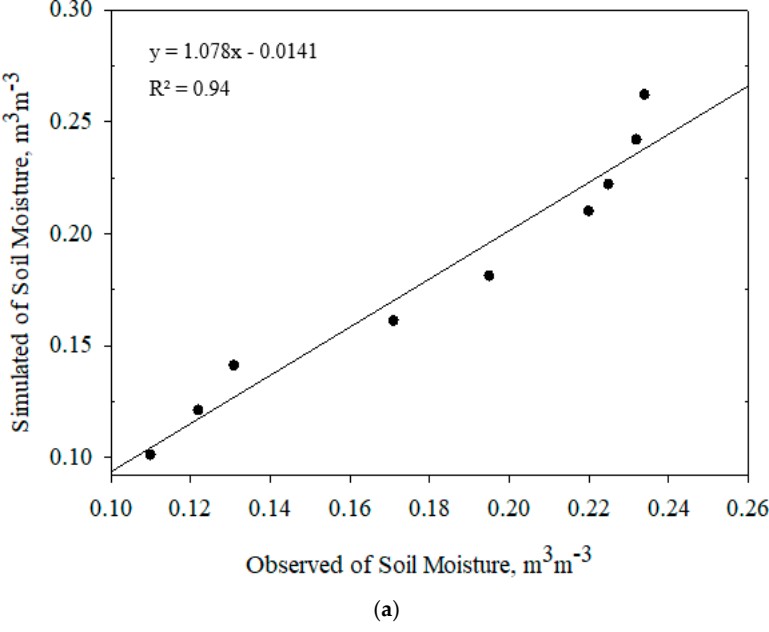

(a)

**Figure 6.** *Cont.*

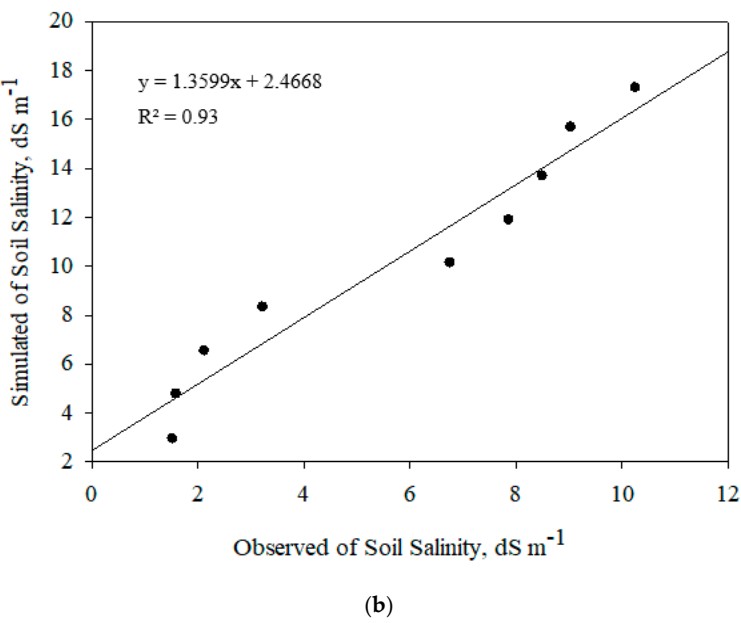

(**b**)

**Figure 6.** Correlation between observed and simulated data for (**a**) soil moisture and (**b**) soil salinity.

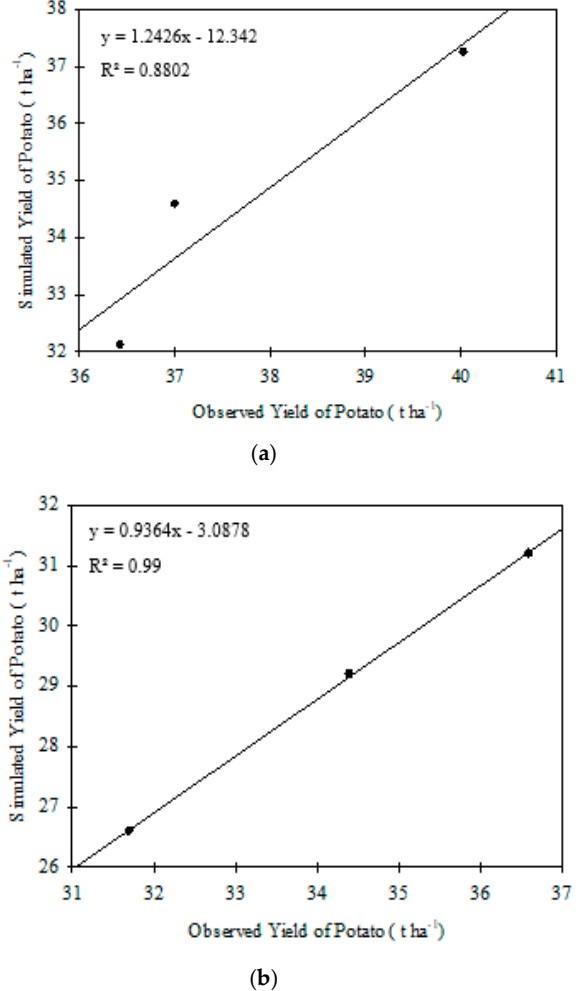

(**a**)

(**b**)

**Figure 7.** *Cont.*

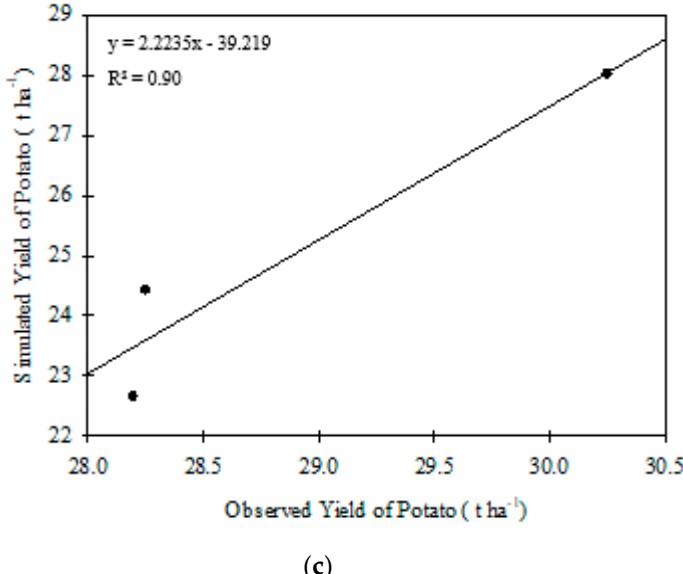

(**c**)

**Figure 7.** Correlation between observed and simulated for potato yield under 150% (**a**), 100% (**b**), and 50% (**c**) of ETc.

## 5. Conclusions

The PRD system resulted in saving water with 75% of ETc which resulted as the same yield under 100% of ETc for both seasons spring and fall of potato crops. In addition, the subsurface irrigation has a better yield compared to surface irrigation. The PRD was more effective in saving water compare to the conventional full or deficit irrigation. The use of PRD resulted in saving water without any significant decrease in yield. Additionally, this study evaluated the SALTMED model's performance and predicted the moisture, salinity of soil, and nitrogen dynamics. The field experiments were conducted in Saudi Arabia using PRD irrigation techniques under irrigation different levels of percentage ETc. The model showed a reasonable outcome for simulating and predicting soil moisture, $R^2$ of more than 90% as an average of all irrigation levels, and salinity distribution with $R^2 = 93\%$. The final potato crop yield had $R^2 = 92\%$ with PRD in arid and semiarid regions. This study used 100% of ETc for the sitting. The calibration of the SALTMED model concentrated on the soil moisture and salinity, which was a good indicator. The SALTMED model confirmed that predicting soil moisture distribution, salinity, and nitrogen was perfect in predicting the final potato yield of potato.

**Author Contributions:** A.A.-O. as supervisor of the work and corresponding author, he wrote the final draft of the manuscript, I.L. is setup the experiments and collect data and supervised all work in the field. A.A. run the saltmed program and contributed on first draft of it. M.H.A.E.-W. contributed in first draft of the manuscript. A.O. collect some of data need it. All authors have read and agreed to the published version of the manuscript.

**Funding:** This research received no external funding.

**Acknowledgments:** The authors extend their appreciation to the Deputyship for Research and Innovation, "Ministry of Education" in Saudi Arabia for funding this research work through the project number IFKSUHI-014.

**Conflicts of Interest:** The author declare no conflict of interest.

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
