# Peer review of "Water Saving and Yield of Potatoes under Partial Root-Zone Drying Drip Irrigation Technique: Field and Modelling Study Using SALTMED Model in Saudi Arabia"

_agronomy, doi:10.3390/agronomy10121997_

Round 1

Reviewer 1 Report

Line

Txt

Suggestions/comments

58

evapotranspiration plant

plant or crop evapotranspiration

68

grain and dry matter yield

70

… the SALTMED model can simulate high relation,

between ?

72

distribution in the soil profile

distribution of what?

75

observer

observed

76

for soil moisture nitrogen dynamics.

for soil moisture and nitrogen dynamics.

78

observer

observation

79, 86

175

679

in Syria by [15] showed

Studies conducted by [16] and [17]

according [23]

In this case, you have to refer Names

87

could be simulated every day

Could make simulations in daily basis?

106

circumstances

use “conditions” instead

125

germination. With

germination ending up to an average

135

the season, fertilizing was stopped

the season,when fertilizing was stopped

135

humic acid was added 6% at a rate of 4 …

Please, rearrange

167-170

Units for irrigation, rainfall and soil moisture

178

weather data

242

seasonality

You mean seasonal reference evapotranspiration or water requirements?

271

distributed more vertically

768

which resulted in same yield of 100% …

… and gave the same yield under the 100% of ETc …

775

reasonable method for simulating

Yoy mean “outcome”?

782

that predicting soil moisture distribution, salinity, and nitrogen was perfect in predicting the final potato yield of potato

Please, rearrange

General suggestion:

Avoid using so many numbers (results) in the text (in the abstract also). The graphs you present give a very good picture of the findings. Please, be more comprehensive in the text by giving only significant numbers and try to make your graphs more visible and uniform (equally dimensioned).

Please, take into account the attached PDF file.

Author Response

Responses to comments and suggestion by reviewer # 1 for the manuscript Agronomy- 1036520

line

text

Comments and suggestions

responses

58

Evapotranspiration plant

Plant or crop evapotranspiration

Changed to crop evapotranspiration

68

Gain and dry matter yield

Changed with thanks

70

The saltmed model can simulate high relation

between

corrected

75

observer

observed

corrected

76

for soil moisture nitrogen dynamics

for soil moisture and nitrogen dynamics.

corrected

78

observer

observation

corrected

79,86

in Syria by [15] showed

done

175

Studies conducted by [16] and [17]

In this case, you have to refer Names

done

679

according [23]

done

87

could be simulated every day

Could make simulations in daily basis?

Done with thanks

106

circumstances

use “conditions” instead

Done with thanks

125

germination. With

germination ending up to an average

Done with thanks

135

the season, fertilizing was stopped

the season, when fertilizing was stopped

Done with thanks

135

humic acid was added 6% at a rate of 4 …

Please, rearrange

Sentence was rewritten

167-130

Units for irrigation, rainfall and soil moisture

Units were added

178

weather data

done

242

seasonality

You mean seasonal reference evapotranspiration or water requirements?

Reference evapotranspiration

271

distributed more vertically

done

768

which resulted in same yield of 100% …

… and gave the same yield under the 100% of ETc …

rewritten

775

reasonable method for simulating

You mean “outcome”?

Changed to outcome

782

that predicting soil moisture distribution, salinity, and nitrogen was perfect in predicting the final potato yield 

Please, rearrange

rewritten

Reviewer 2 Report

The manuscript “Water Saving and Yield of Potatoes Using Partial Root-Zone Drying Drip Irrigation Technique: Field and Modelling Study Using SALTMED Model” seems to be an excellent topic and time-worthy work indeed. Using sensors can save valuable time and reduce the expenses related to the estimation of irrigation water in drought-prone and desert areas. The model used in this study will be a useful tool for agricultural practice in such a place where the water scarcity problem is severe. However, some queries need to address to improve the manuscript. Some technical information was incorrectly written along with typo errors in the whole manuscript. I strongly suggest revising and check the entire manuscript carefully, for which I recommend minor revision.

Line 58-60: Please check the sentence carefully, did you mean plant water uptake?

Line 66-69: Please check the sentence carefully, did you mean grain dry matter yield?

Line 238-241: The result explained here is confusing. What does it mean by all season? Table 5 comparing the spring and fall seasons only. We can see in both season Penman-Monteith showed higher result than Pan evaporation.

Line 249-255: Too much confusing result explained here. I did not find the relevant result from table 6b. I think “At PRD-SS yield decrease by 5.1% compare to PRD-S treatment” statement indicating table 6a.  Yield increased in CDI-SS by 2.6% compare to CDI-S, or others should mention. Table number and season is not written correctively. Potato yield results should rewrite according to the table's data and mention the table number accordingly.

Line 263: I did not find soil salinity and nitrogen data in figure 3a.

The stage should specify by using a,b,c, and d as like Figure 3b in all cases.

Line 378: Figure 4a is not showing any data regarding salinity and nitrogen.

Line 675-682: To be honest, I could not understand this discussion part that the authors explained in table 5. I would suggest checking the information given here is correct or not.

Line 738-739: Table 4 is not showing any data regarding statistical indicators. Please check whether it will be Table 4 or Table 7.

Line 755: Figure must be specified by a, b, c, etc., as mentioned in results and discussions for all cases.

Author Response

Comments and suggestion by reviewer # 2 for the manuscript 1036520

Comment: The manuscript “Water Saving and Yield of Potatoes Using Partial Root-Zone Drying Drip Irrigation Technique: Field and Modelling Study Using SALTMED Model” seems to be an excellent topic and time-worthy work indeed. Using sensors can save valuable time and reduce the expenses related to the estimation of irrigation water in drought-prone and desert areas. The model used in this study will be a useful tool for agricultural practice in such a place where the water scarcity problem is severe. However, some queries need to address to improve the manuscript. Some technical information was incorrectly written along with typo errors in the whole manuscript. I strongly suggest revising and check the entire manuscript carefully, for which I recommend minor revision.

Response: Thank you.

Comment: Line 58-60: Please check the sentence carefully, did you mean plant water uptake?

Response: The sentence was checked and rewritten.

Comment: Line 66-69: Please check the sentence carefully, did you mean grain dry matter yield?

Response: corrected.

Comment: Line 238-241: The result explained here is confusing. What does it mean by all season? Table 5 comparing the spring and fall seasons only. We can see in both season Penman-Monteith showed higher result than Pan evaporation.

Comment: the sentence was rewritten.

Comment: Line 249-255: Too much confusing result explained here. I did not find the relevant result from table 6b. I think “At PRD-SS yield decrease by 5.1% compare to PRD-S treatment” statement indicating table 6a.  Yield increased in CDI-SS by 2.6% compare to CDI-S, or others should mention. Table number and season is not written correctively. Potato yield results should rewrite according to the table's data and mention the table number accordingly.

Response: Thank you. The paragraph was rewritten.

Comment: Line 263: I did not find soil salinity and nitrogen data in figure 3a.

Response: nitrogen data was removed from the sentence. This happened as we separated the figures

Comment: The stage should specify by using a,b,c, and d as like Figure 3b in all cases.

Response: Thank you done and added to figures.

Comment: Line 378: Figure 4a is not showing any data regarding salinity and nitrogen.

Response: corrected as mentioned before.

Comment: Line 675-682: To be honest, I could not understand this discussion part that the authors explained in table 5. I would suggest checking the information given here is correct or not.

Response: the information was checked and corrected.

Comment: Line 738-739: Table 4 is not showing any data regarding statistical indicators. Please check whether it will be Table 4 or Table 7.

Response: it was corrected to table 7. Tlhanks

Comment: Line 755: Figure must be specified by a, b, c, etc., as mentioned in results and discussions for all cases.

Response: a, b, and c added to the figures.
